# Effects of Fluctuating Thermal Regimes on Life History Parameters and Body Size of *Ophraella communa*

**DOI:** 10.3390/insects13090821

**Published:** 2022-09-09

**Authors:** Chenchen Zhao, Hongsong Chen, Jianying Guo, Zhongshi Zhou

**Affiliations:** 1State Key Laboratory for Biology of Plant Diseases and Insect Pests, Institute of Plant Protection, Chinese Academy of Agricultural Sciences, Beijing 100193, China; 2Guangxi Key Laboratory for Biology of Crop Diseases and Insect Pests, Institute of Plant Protection, Guangxi Academy of Agricultural Sciences, Nanning 530007, China; 3International Laboratory for Green Pest Control, College of Plant Protection, Henan Agricultural University, Zhengzhou 450002, China

**Keywords:** leaf beetle, common ragweed, thermal physiology, population fitness

## Abstract

**Simple Summary:**

The ragweed is a global problem with its wide distribution and substantial economic and ecological cost. The ragweed beetle is an effective biological control agent for ragweed. With the increasing challenges of climate change, understanding the population responses of the ragweed beetle to thermal stress is essential to predict the impacts of climate change more accurately. Our study showed that *Ophraella communa* has stronger capabilities to adjust the development time with prolonged adult longevity, regulating sex differentiation and response to fluctuating temperatures. Taken together, the adaptation strategies of the ragweed beetle make it play a crucial role as an effective biological control agent for ragweed.

**Abstract:**

The beetle *Ophraella communa* is an effective biological control agent against the invasive common ragweed spread across various ecosystems with variable temperature ranges. The trend in climate change attributed to fluctuating temperatures and abrupt rainfalls is expected to continue. This study aimed to better understand the effects of thermal fluctuation on *O. communa* by exposing all their life stages to heat stress under different treatments. Repeated exposure to high temperatures, relative to constant milder temperatures, increased the duration of immature development, mean generation time, and the adult longevity, decreased the intrinsic rate of increase, finite rate of population increase, net reproductive rate, survival rate, overall longevity, body length, and mass of adults and positively affected overall fecundity by prolonging the oviposition period, biasing sex ratio towards females. After exposure to heat stress, the mating success and production of viable offspring were higher in *O. communa*. Our findings demonstrate that exposure to heat stress negatively affects ragweed beetles, but they were able to survive and reproduce.

## 1. Introduction

The common ragweed, *Ambrosia artemisiifolia* L. (Asterales: Asteraceae), creates serious economic, ecological, and health problems as a key cause of allergic rhinitis and asthma [1,2]. This plant is notoriously invasive due to its strong competitive ability and the release of large amounts of pollen, spreading worldwide from North America to continents as far as Europe, Eurasia, Australia, Africa, and Asia [3]. In China, ragweed has rapidly increased in distribution range and abundance after invading and establishing itself in more than 22 provinces across the country [4].

The ragweed leaf beetle, *Ophraella communa* LeSage (Coleoptera: Chrysomelidae), is also indigenous to North America and has been spread worldwide and recognized as an effective specific biological control agent against *A. artemisiifolia* [5]. Additionally, as known that *O. communa* is an oligotrophic insect credited with delivering better control of common ragweed than typical herbicide applications [6]. There is increasing evidence that the international spread of the common ragweed may have been significantly hampered by the presence of this leaf beetle [6,7,8]. 

Temperature is a key factor driving species distributions, especially in ectothermic organisms [9]. In China, *O. communa* has spread along with *A. artemisiifolia* across various cold and hot ecosystems [4]. The adaptations of *O. communa* to low temperatures have been well studied, with the main focus on how cold-hardiness allows *O. communa* to overwinter [4,10]. Effects of higher temperatures on the leaf beetle development and growth have not yet been adequately studied, particularly the effectiveness of biological control against *A. artemisifolia* under the increasing temperatures associated with climate change [11]. A previous study indicated that climate change might exacerbate the risk of plant invasions [12]. Understanding the populational responses of *O. communa* to thermal stress is essential to more accurately predict the impacts of climate change on the distribution, population phenology, and, ultimately, the long-term potential of this species as a biological control agent.

In this context, we investigated how fluctuating thermal regimes influence the life-history parameters of *O. communa*, specifically the pre-adult development stage, survival rates, adult fecundity, longevity, body mass and length, adult oviposition periods, and the hatching rate.

## 2. Materials and Methods

### 2.1. Host Plants

*A. artemisiifolia* seeds were collected, stored, and grown as described in previous studies [13,14]. When the plants reached heights between 30 and 50 cm, they were transplanted to individual pots (diameter 21 cm and height 17 cm) to be presented as hosts for *O. communa* in further experiments.

### 2.2. Insect Culture

The adults of *O. communa* were collected and reared (Appendix A) as described in the previous study [15]. Adults were categorized based on an inverted ‘V’ stripe at the end of the male’s genital abdominal segment, lacking in females [16]. Thirty adult couples of *O. communa* (F_0_) were randomly obtained from a laboratory colony and transferred onto 10 freshly potted ragweed plants (random distribution, beetles can fly) and caged within nylon gauze bags (40 mesh size, 2 m × 2 m × 2 m) in a green-house for oviposition. The adult beetles were removed two days later, and all plants containing eggs (F_1_) were reserved for the next experiment. This procedure was repeated five times (150 couples and 50 ragweed plants for each temperature treatment were used to produce enough eggs for the next experiment).

### 2.3. Fluctuating Thermal Regimes

The duration and intensity of heat stress treatments were based on the recorded duration and intensity of peak summer temperatures in central China, i.e., maximum temperature around 44 °C for approximately 3 h every day for 3–5 consecutive days [17,18] (Figure 1C,D). For all of the experiments, the insects were held at the constant temperature of 28 °C with experimental groups of eggs, larvae, pupae, and adults being exposed for 3 h (from 12:00 to 15:00) to either 40, 42, or 44 °C, while the control group was maintained at 28 °C. The regulators maintain the temperature in environmental chambers without moving the insects. The eggs were exposed to elevated temperatures for three consecutive days, while all the other life stages (larvae, pupae, and adults) were exposed for 5 consecutive days [19]. These periods were decided based on the recorded duration of the pre-adult developmental stage of *O. communa* when reared at the constant high temperature of 32 °C: 4.0 days for eggs, 7.6 days for larvae, and 6.0 days for pupae [20].

Experimental treatments were performed in separate environmental chambers (PRX-450D, Ningbo Haishu Safe Experimental Equipment Co., Ltd., Zhejiang, China) with temperatures respectively set to 28, 40, 42, or 44 °C (±1 °C) and relative humidity at 70 ± 5% under a photoperiod of 14:10 (L:D) h, with the light intensity of 12,000 lx in each chamber.

Potted plants containing eggs were placed in plastic basins (50 cm × 30 cm) and randomly allocated into the environmental chambers. An approximate total of 1000 eggs (28 °C: 936 eggs: 40 °C: 3020 eggs; 42 °C: 1575 eggs, and 44 °C: 2311 eggs; we ensured each potted fresh common ragweed plant has about 60 eggs, and 20–50 plants with eggs were used in each treatment) were thus selected for exposure to each of the experimental temperatures (28 °C, 40 °C, 42 °C, or 44 °C, while 3–10 environmental chambers were used in each treatment) until the emergence of adults [17].

Eggs (F_1_) were checked and recorded daily for hatching. As most freshly laid eggs in the same temperature treatment may hatch within the same day, we were able to track the development time in days for each larva based on their growth until pupation (the neonates of one egg mass were kept on the same common ragweed plant until pupation). Larvae were retained together on their original potted plants until pupation. Pupae were detached and placed individually in unsealed cuvettes to be checked daily until adult emergence [20].

The resulting F_1_ generation adults were distributed in couples onto freshly potted ragweed. These were then exposed to high temperatures in the environmental chambers as previously described (3 h daily for 5 consecutive days), after which the infested plants were moved to a greenhouse (temperature: 28 ± 2 °C; with natural light conditions and relative humidity of 70 ± 5%).

Newly emerged adults were grouped based on sex, weighed with an electronic precision balance (to within 0.01 mm) [21], and measured (to within 0.01 mm) from the front to the tip of the abdomen under a stereomicroscope, after which couples were allowed to mate. Newly emerged *O. communa* adults at different treatments were mated. Each pair was placed on a fresh twig of common ragweed, which was inserted into a plastic bottle (3 cm in diameter and 5 cm in height) with a hole (0.8 cm in diameter). The bottles were filled with water and placed in transparent plastic boxes (19 cm × 12 cm × 6 cm) covered with mesh to prevent adult escape. Fresh twigs were changed daily, and the eggs (F_2_) were counted [22]. The longevity and fecundity (number of eggs laid), adult pre-oviposition period (APOP), and total pre-oviposition period (TPOP) of the F_1_ beetles subjected to each treatment were recorded daily. After that, the proportion of hatched eggs (F_2_) was recorded.

### 2.4. Data Analysis

Data were analyzed as a two-sex life table according to with TWO-SEX-MSChart program. The age-stage specific survival rate (*s_xj_*) and the net reproductive rate (*R_0_*) (offspring/individual) were calculated [23]. The computational formulae of life table parameters used were:(1)$$lx=\sum_{j=1}^{m}s_{xj}$$
(2)$$m_{x}=\frac{\sum_{j=1}^{m}s_{xj}f_{xj}}{\sum_{j=0}^{m}s_{xj}}$$
(3)$$R_{0}=\sum l_{x}m_{x}$$
(4)$$\sum_{x=0}e^{-r(x+1)}l_{x}m_{x}=1$$
(5)$$T=\frac{\ln R_{0}}{r}$$
where *m* is the number of developmental stages, *x* is the age, *l_x_* is the age-specific survival rate, *m_x_* is age-specific fecundity, *T* is the mean generation time, *R_0_* is the net reproductive rate, *r* is the intrinsic rate of increase estimated using the Euler–Lotka formula (Equation (2)) with age indexed from 0, and λ is the finite rate of increase.

Experiments were designed to estimate the effects of temperature exposure on the population-scale fitness of *O. communa* populations. Towards estimations, we used the bootstrap technique [24] with 5000 bootstraps to generate stable estimates for the variances and standard errors of the age-specific survival rate, longevity, fecundity, adult pre-oviposition period (APOP), the total pre-oviposition period (TPOP), and population parameters. The difference between treatments was statistically compared to the 5% significance level through a paired bootstrap test. The bootstrap method and paired bootstrap test were embedded into the TWOSEX-MSChart computer program. The TIMING-MSChart software (available from the same website as TWOSEX-MSChart [25]) was used to project the population growth rates for 60 days. Significant differences in the body length, body mass, and egg hatchability were determined by the LSD test (*p* < 0.05, SAS Institute, 2004).

## 3. Results

### 3.1. Population Development of Ophraella communa under Thermal Fluctuation

Significant effects on the development duration of *O. communa* were observed under different temperatures (Table 1). The egg and pupal developmental time of *O. communa* significantly increased under higher temperature exposure (Table 1). The duration of the larval development recorded for *O. communa* at 40 °C was significantly longer (11.33 days) than under other temperatures (7.34 d, 7.58 d, and 7.31 d), which were not significantly different.

Adults reared at 28 °C emerged significantly earlier (i.e., had faster pre-adult development) and produced eggs earlier (APOP and TPOP) than others reared under higher temperatures. Overall, longevity decreased significantly with exposure to increasing temperatures (Table 1). However, those beetles that survived as adults under 40 and 42 °C lived longer and presented longer-lasting oviposition periods, leading to a higher fecundity rate than other beetles under 28 °C (Table 2). Interestingly, when the rearing temperature reached 44 °C, beetles displayed similar oviposition periods with less fecundity as compared to 28 °C (Table 2).

### 3.2. Age-Specific Survival Rate and Population Parameters of Ophraella communa under Thermal Fluctuation

Significant differences were observed in the survival rates of beetles reared at different temperature treatments (Table 3), as represented by *S_xj_* curves (Figure 2). Thermal fluctuation significantly reduced age/stage-specific survival rates (egg, larva, and adult). Eggs proved heat resistant regardless of exposure temperatures (for instance, over 55% of eggs were viable after exposure to the highest temperature). The pupal survival rate was unaffected by rearing temperatures under 40 °C but significantly reduced under exposure to 42 °C and 44 °C. Furthermore, the sex ratio of females to males was significantly different among the four temperatures, where the proportion of females was more significant increased under higher rearing temperatures (Table 3).

### 3.3. Population Dynamics of Ophraella communa under Thermal Fluctuation

Figure 3 shows the age-specific survival rate (*l_x_*), female age-specific fecundity (*f_x_*_4_; the female adult being the fourth life stage), age-specific fecundity (*m_x_*), and age-specific net fecundity (*l_x_m_x_*) of *O. communa*. The *l_x_* curves of *O. communa* increased with increasing longevity. In contrast, *m_x_* and *l_x_m_x_* showed a decreasing trend relative to age and rearing temperature (Figure 3). A higher daily *m_x_* was observed in females exposed to 44 ºC (*m_x_* = 27 eggs per day up to the age of 95 d). In contrast, daily *m_x_* values of other females reached 13 eggs per day at the ages of 40 d at 40 °C, 27d at 28 °C, and 38d at 42 °C, respectively. By factoring in the survival rate under increasing temperatures, we obtained the highest *l_x_m_x_* values of *O. communa* as 8.7 (28 °C), 4.4 (40 °C), 1.5 (42 °C), and 0.6 (44 °C) offspring, respectively.

The age-stage life expectancy (*e_xj_*) of *O. communa* shows the total time that an individual of age x and stage j is expected to live (Figure 4.). Life expectancy for newly laid eggs (*e*_01_) was 52.87 d (28 °C), 36.78 d (40 °C), 18.82 d (42 °C), and 15.43 d (44 °C), respectively. These values were proportional to the mean longevity of all individuals, according to Table 1. Due to the high mortality during the egg–larva stage, the estimated life expectancy of the egg–larva stage was initially lower and later increased. The maximum life expectancies occurred at: age 9 days under 28 °C (*e_xj_* = 69.24), age 11 days under 40 °C (*e_xj_* = 96.73), age 14 days under 42 °C (*e_xj_* = 86.11), and age 13 days under 44 °C (*e_xj_*= 60.19), as calculated for male beetles (Figure 4).

The reproductive value (*ν_xj_*) is the expected contribution of a given individual of age *x* and life stage *j* to a projected future population. The reproductive value increased when adult females emerged and at the onset of reproduction. Major reproductive curve peaks were observed for *O. communa* at the ages of 24 days (*v*_24,4_ = 141 in 28 °C), 33 days (*v*_33,4_ = 212 in 40 °C), 32 days (*v*_32,4_ = 222 in 40 °C), and 70 days (*v*_32,4_ = 259 in 44 °C) (Figure 5). Peak reproductive values of *O. communa* following exposure to 44 ºC were the highest and longest among all treatments.

The intrinsic increase rates, the finite rates, and the net reproductive rates of *O. communa* significantly decreased with exposure to increasing temperatures. The highest intrinsic rate of increase (*r* = 0.1812 d^−1^), finite rate of increase (*λ* = 1.199 d^−1^), and net reproductive rate (*R*_0_ = 212.50 offspring) were all observed at 28 °C, whereas the lowest rates were observed from exposure to 44 °C. On the other hand, mean generation times (*T*) of *O. communa* also increased with the exposure temperatures (Table 4).

### 3.4. Population Projection

Figure 6 presents projected population growths using obtained parameters for each treatment, assuming an initial population of 10 eggs and the emergence times for each life stage. Changes to the stage structure could be easily depicted using a population projection extrapolated from the age-stage, two-sex life table data. The projection shows that the growth rates of *O. communa* decrease with exposure to higher temperatures, but the population should be able to sustain two generations after 60 days from exposure to 44 °C.

### 3.5. Adult Fitness of Ophraella communa

Heat stress significantly affected the body growth of *O. communa*; individuals reared under higher temperatures had smaller bodies compared to those reared at 28 °C (length, female: *F*_3,116_ = 29.53, male: *F*_3,116_ = 27.54; mass, female: *F*_3,116_ = 27.59, male: *F*_3,116_ = 9.79) (*p* < 0.001, Figure 7A,B). Females were always significantly larger than males (body length; *p* < 0.001, Figure 7B).

Thermal fluctuation also significantly reduced the hatchability of F_2_ eggs from rearing temperatures above 40 °C (*F*_3,80_ = 10.32, *p* < 0.001) (Figure 7C). Compared to F_1_, the egg hatchability of F_2_ was significantly higher (Table 3, Figure 7C).

## 4. Discussion

Heat tolerance in insects is weakly linked to latitude, but the most tolerant species often live in warm microhabitats [26]. Climate change may exacerbate the risk of plant invasions [12]. Global warming likely aggravates the problem of the invasive ragweed [27], which can rapidly evolve in response to climate change within a single generation [28], given that it has invaded and established itself mostly in warm regions in China (Figure 1) [4,29]. High-temperature tolerance of natural enemies thus becomes paramount for efficient prevention and control of ragweed. Ambient temperature profoundly influences the population distribution, life history, behavior, and species abundance of insects [10]. On the other hand, insects have a variety of biochemical and behavioral mechanisms to avert damage from exposure to extreme temperatures [30,31]. Such mechanisms manifest during long-term (e.g., seasonal acclimation) and short-term exposure to altered temperatures [32]. However, predicting the impacts of heatwaves on insects at population level can be challenging [33]. In this study, we have documented several effects of fluctuating thermal regimes (40, 42, and 44 °C) on the population fitness of *O. communa*. Overall, the results suggest that the induced exposure to heat stress negatively affects ragweed beetles, but they are able to survive and reproduce.

Heat stress significantly decreased the survival rates of the pre-adult developmental stages of *O. communa* and their respective hatching, pupation, and emergence rates. Pupal survival was the only exception, remaining unaffected by exposure to 40 °C. These results illustrate that the physiological responses to heat stress can significantly vary across the life stages of a given insect. A previous study demonstrated that the heat resistance of insects tends to decrease with the development stage [34]. It seems noteworthy that the eggs of *O. communa* proved resistant to heat, regardless of exposure temperature. There is yet no clear pattern for the survival rates of the different life stages of insects under heat stress [35,36], as the sensitivity of different insect species to heat stress seems to vary according to the experimental conditions.

Studies have shown that extreme temperatures can greatly reduce the survival and reproductive success of insects, suggesting a dramatic impact of extreme weather events on biodiversity [37]. In contrast, our results showed that thermal fluctuation could stimulate the fecundity of beetle females, illustrating some apparent adaptiveness of *O. communa* to thermal fluctuation that could improve egg hatchability from the initial heat exposure of F_1_ individuals.

Exposure to high temperatures increased the pre-adult developmental duration of *O. communa*. Thermal fluctuations can often allow insect development outside the temperature range [38,39]. However, potentially lethal temperatures generally will delay development relative to optimal temperatures [39,40]. Such delays are likely a consequence of direct heat damage and associated physiological costs for biochemical repair [41,42].

Thermal fluctuation can often extend insect longevity. A previous study has demonstrated a positive relationship between increasing temperatures and insect longevity [43]. Similarly, the longevity of adult *O. communa* increased under exposure to higher temperatures. Fluctuating temperatures have been reported to increase [44], decrease [45], or cause no effect on the life span of insects [46]. A straightforward trade-off between damage repair and somatic maintenance could reduce longevity if some injury is caused by brief exposure to higher or lower temperatures [47]. Artificial exposure to 40 °C and 42 °C significantly increased the longevity of *O. communa* adults, but exposure to 44 °C has reduced longevity, possibly illustrating some adaptive response of the ragweed beetle to heat stress that may improve their survival [48]. Overall, the results indicated that temperatures above a certain threshold would increase physiological effects affecting the survival rate of *O. communa*.

High temperature is described to reduce the survival rate of insects and can reduce or even suppress their fecundity [49]. However, in contrast to such expectation, a prolonged oviposition period in female ragweed beetles was observed after exposure to high temperatures, driving a significant increase in overall fecundity. Furthermore, previous studies have demonstrated a positive relationship between female fecundity and adult body size [50,51]. In the present study, exposure to higher temperatures yielded smaller adult body size, which contradicts the assumptions about different insect species.

Whether an individual does or does not survive acute heat stress, its ability to subsequently mate and produce viable offspring is crucial to sustaining the population [37]. Temperatures considered stressful to insects usually impair oocyte development [52], decreasing mating success [53], sperm production, and sperm viability [54], leading to lower reproductive. However, the present investigation demonstrated that thermal fluctuation could lead to a prolonged oviposition period in the ragweed beetle, whose eggs mostly hatched into larvae even under the highest tested temperature.

The survival of F_0_ individuals after stressful heat exposure demonstrates that ragweed beetles are relatively heat resistant. Some physiological adaptiveness of *O. communa* to heat stress might improve egg hatchability after a generation gets exposed to heat stress. This may have led to the fact that enough *O. communa* larvae hatched, allowing F_1_ to thrive under the imposed hot conditions. Concomitantly, exposure to high temperatures also increased the female ratio of *O. communa*. It has been postulated that resilience to climate warming may result from increased fecundity of females following a decline in offspring survival from heat stress [55]. This would make for an intriguing phenomenon where *O. communa* modulates its sex ratio as an adaptive response to heat stress, which would benefit the biological control of the ragweed [6]. Temperature-dependent sex determination is widespread among reptiles (e.g., crocodilians, turtles [56]), where males are produced under mild temperatures and females are chiefly produced under extremes of heat and cold (e.g., *Alligator mississippiensis* [57], *Lepidochelys olivacea* [58]). Temperature-dependent sex determination is a positive response of some species to rapid global climate changes [59,60], warranting further study.

Body size plays a crucial role in insect survival against adversity. Temperature-dependent plasticity of insect body size adapts to environmental variations [61]. Normally, increased temperatures during development will result in smaller individuals because accelerated metabolism leads to faster development [62,63]. This temperature–size relationship arises due to phenotypic plasticity or selection for smaller body sizes [64]. In our study, body size was affected by temperature in a complex manner: thermal fluctuation significantly reduced the body length in *O. communa* after delaying development, which conforms to Bergmann’s rule and the temperature–size rule (i.e., hotter conditions generate smaller adults) [65]. Indeed, the body size is usually recorded as inversely correlated with rearing temperature in insects [66,67]. Within certain physiological limits, higher temperatures will accelerate the development and growth of insects [32]. It should be noted that smaller insects gain and lose heat more quickly than larger insects [68], and they also accumulate less heat, which turns advantageous for their survival in the hottest regions at low latitudes [68]. Males of *O. communa* are typically smaller than females at any given temperature because of sexual dimorphism [69,70]. A previous study shows that females are larger than males in 88% of insect species, while only 7% of insect species in males are bigger than females, implying that smaller adult males are likely more heat tolerant than females [71]. Taken together, these results illustrate how the physiological response to thermal fluctuation may vary according to the life stage and sex of *O. communa*.

Klockmann et al. (2017) [37] stated that any predictions regarding the fate of a given species under changing environmental conditions should consider the thermal tolerance throughout their ontogeny, body mass range, and acclimation responses. In this study, we demonstrated that thermal fluctuation could significantly reduce the body mass of *O. communa*. Similarly, Karan et al. (1998) showed that *Drosophila melanogaster* were significantly lighter under increasing rearing temperatures [72]. Additionally, high-amplitude temperature fluctuations reduced the pupal size of *Manduca sexta* [73] and the wing size and body mass of *D. melanogaster* and *D. simulans* [74,75,76].

Adaptive evolution over a short timescale is well documented in common ragweed and response to climate change [77]. Under high temperatures, *O. communa* may have stronger capabilities in adjusting the development time, prolonging adult longevity, regulating sex differentiation, and responding to global warming, which allows them to play their crucial role as a natural enemy. Our results showed that even exposure to heat stress negatively affects ragweed beetles, but they were able to survive and reproduce. These findings help understand *O. communa* ecological behavior and improve its utilization as a control agent, for instance, by adjusting the release of *O. communa* (e.g., by life stages and sex ratio) in nature to enhance the impacts on its host plant *A. artemisiifolia*.

## Figures and Tables

**Figure 1 insects-13-00821-f001:**
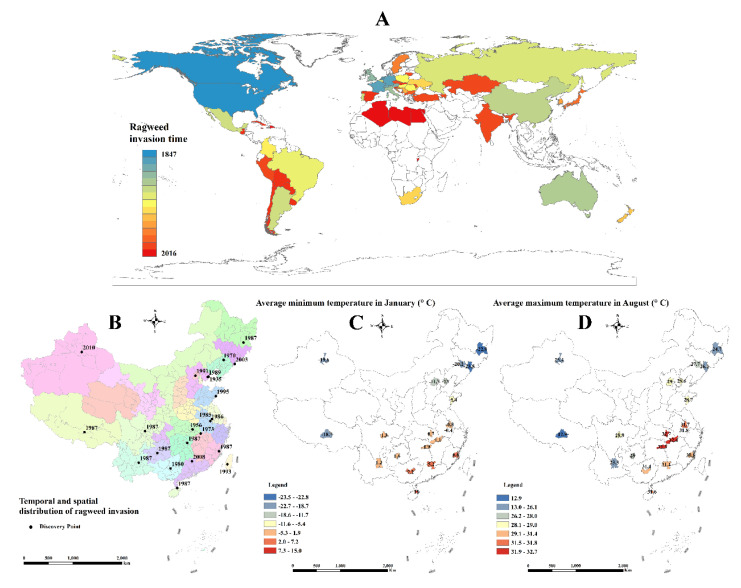
The geographical distribution and time scale of the invasion of the common ragweed in the world (**A**) and in China (**B**), and the mean low (**C**) and high temperatures (**D**) of the invaded habitats in China (2010–2018) [17,18].

**Figure 2 insects-13-00821-f002:**
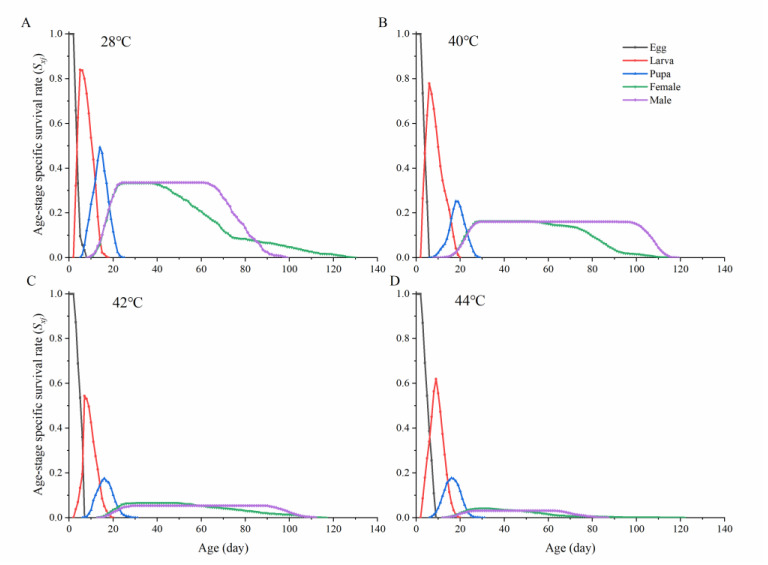
The age-specific survival rate (*s_xj_*) of *Ophraella communa* exposed to different stressful high temperatures. (**A**): 28 °C, (**B**): 40 °C, (**C**): 42 °C, (**D**): 44 °C.

**Figure 3 insects-13-00821-f003:**
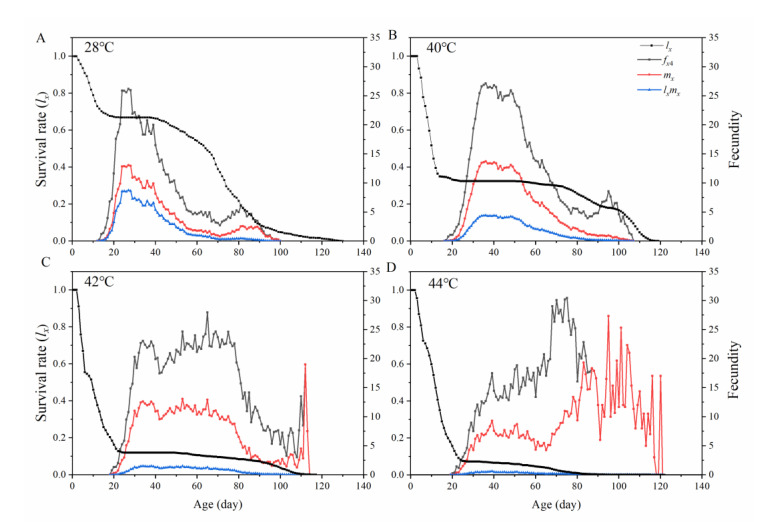
The age-specific survival rate (*l_x_*), age-stage specific fecundity (*f_x_*_4_), age-specific fecundity (*m_x_*), and age-specific net fecundity (*l_x_m_x_*) of *Ophraella communa* at different heat stress. (**A**): 28 °C, (**B**): 40 °C, (**C**): 42 °C, (**D**): 44 °C.

**Figure 4 insects-13-00821-f004:**
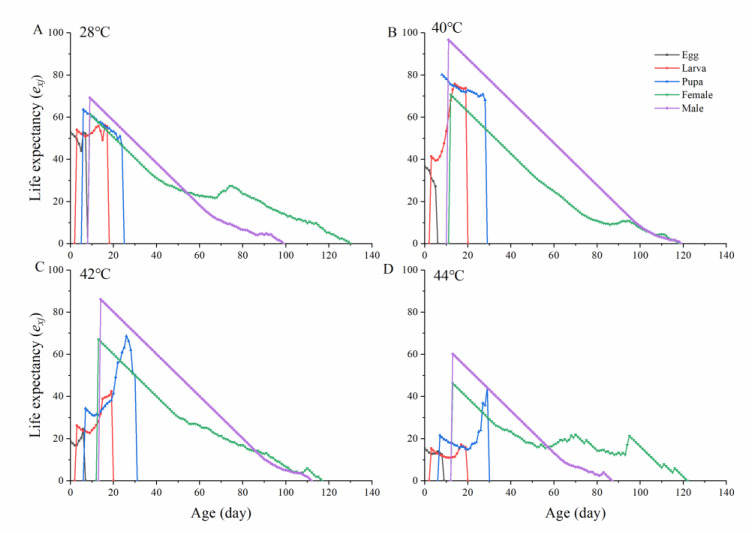
The age-specific life expectancies (*e_xj_*) of *Ophraella communa* reared under different temperatures. (**A**): 28 °C, (**B**): 40 °C, (**C**): 42 °C, (**D**): 44 °C.

**Figure 5 insects-13-00821-f005:**
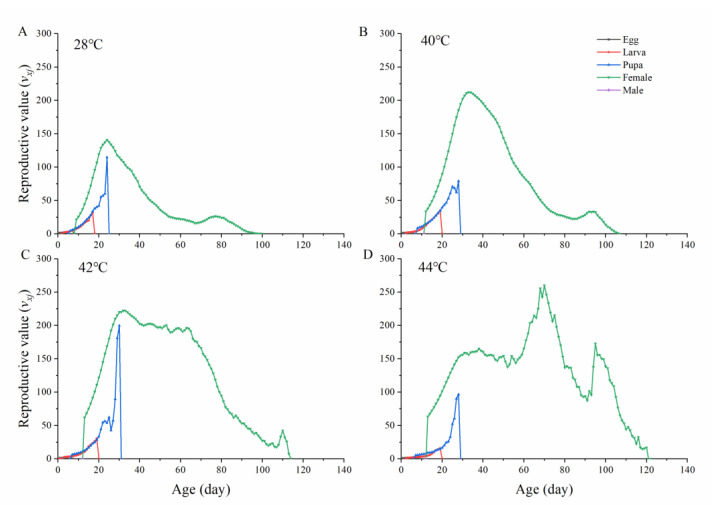
Age-specific reproductive values (*v_xj_*) of *Ophraella communa* reared under different temperatures. (**A**): 28 °C, (**B**): 40 °C, (**C**): 42 °C, (**D**): 44 °C.

**Figure 6 insects-13-00821-f006:**
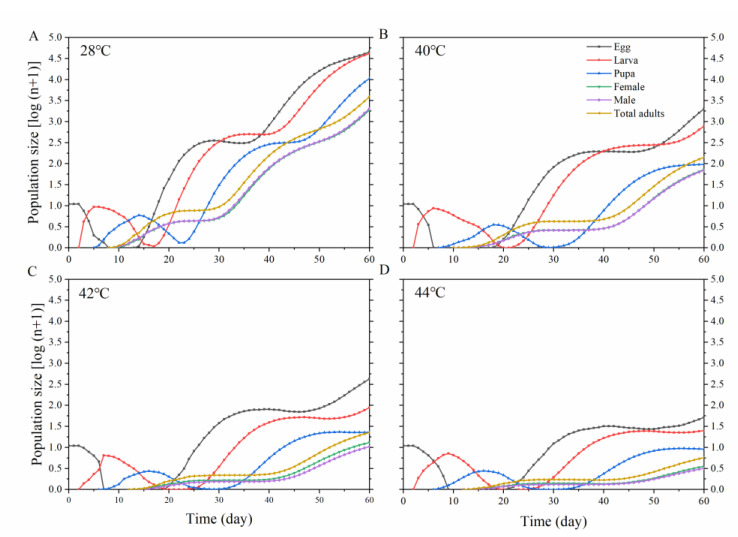
Projection of *Ophraella communa* population growth reared under different temperatures considering an initial population of 10 eggs. (**A**): 28 °C, (**B**): 40 °C, (**C**): 42 °C, (**D**): 44 °C.

**Figure 7 insects-13-00821-f007:**
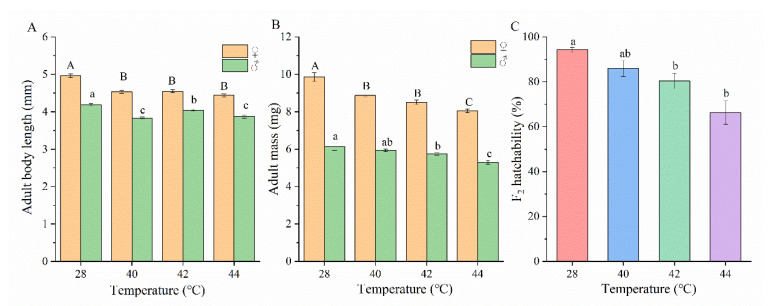
Effects of different temperatures on the adult fitness of the ragweed beetle *Ophraella communa*. (**A**): adult length, (**B**): adult mass, (**C**): F_2_ egg hatchability. Mean ± SE; different lowercase or capital letter are statistically different at *p* < 0.05 level of the same sex in the same fitness. Experimental temperatures of 40, 42, and 44 ℃ were imposed on different stages of *O. communa* in environmental chambers from 12:00–15:00 daily until a generation was completed. A standard temperature of 28 ℃ was otherwise used for other life stages and the control group.

**Table 1 insects-13-00821-t001:** Effects of periodical exposure to high temperatures on the life cycle of the ragweed beetle *Ophraella communa*.

Population Parameters	Developmental Period and Longevity (Days)			
	28 °C (*n*)	40 °C (*n*)	42 °C (*n*)	44 °C (*n*)
Egg development duration	4.15 ± 0.04 ^a^ (936)	4.29 ± 2.33 ^b^ (3020)	6.21 ± 0.04 ^c^ (1575)	6.30 ± 0.05 ^c^ (2311)
Larva development duration	7.34 ± 0.09 ^a^ (668)	11.33 ± 0.07 ^b^ (1060)	7.58 ± 0.12 ^a^ (459)	7.31 ± 0.09 ^a^ (660)
Pupa development duration	5.76 ± 0.08 ^a^ (626)	6.36± 0.07 ^b^ (980)	6.80 ± 0.14 ^c^ (188)	6.68 ± 0.16 ^bc^ (168)
Pre-adult development duration	17.22 ± 0.13 ^a^ (626)	22.01± 0.11 ^c^ (980)	20.79 ± 0.26 ^b^ (188)	20.35 ± 0.27 ^b^ (168)
Adult development duration	57.21 ± 0.66 ^b^ (626)	73.07 ± 0.49 ^c^ (980)	68.33 ± 1.22 ^d^ (188)	44.97 ± 1.20 ^a^ (168)
Female adult development duration	53.28 ± 1.24 ^b^ (312)	60.76 ± 0.58 ^c^ (495)	59.68 ± 1.81 ^c^ (103)	38.85 ± 1.87 ^a^ (95)
Male adult development duration	61.11 ± 0.43 ^b^ (314)	85.64 ± 0.14 ^d^ (485)	78.82 ± 0.46 ^c^ (85)	52.93 ± 0.59 ^a^ (73)
Overall longevity ^1^	52.87 ± 1.12 ^a^ (936)	36.78 ± 0.76 ^b^ (3020)	18.82 ± 0.68 ^c^ (1575)	15.43 ± 0.33 ^d^ (2311)
Longevity ^2^	74.43 ± 0.68 ^b^ (626)	95.08 ± 0.51 ^d^ (980)	89.13 ± 1.25 ^c^ (188)	65.32 ± 1.22 ^a^ (168)
Longevity (female adult)	70.59 ± 1.26 ^b^ (312)	82.69 ± 0.59 ^c^ (495)	80.07 ± 1.82 ^c^ (103)	59.24 ± 1.89 ^a^ (95)
Longevity (male adult)	78.25 ± 0.46 ^b^ (314)	107.73 ± 0.20 ^d^ (485)	100.11 ± 0.61 ^c^ (85)	73.20 ± 0.69 ^a^ (73)
APOP	4.80 ± 0.04 ^a^ (312)	5.76± 0.05 ^b^ (495)	6.54 ± 0.08 ^c^ (103)	8.02 ± 0.12 ^d^ (95)
TPOP	22.11 ± 0.18 ^a^ (312)	27.69 ± 0.15 ^bc^ (495)	26.93 ± 0.36 ^b^ (103)	28.43 ± 0.39 ^c^ (95)

Overall longevity ^1^: average longevity of all beetles, Longevity ^2^: average longevity of the beetles which experienced egg stage, larval period, pupal period, and survived to adult. APOP: adult pre-oviposition period, TPOP: the total pre-oviposition period. The number of beetles is given in parentheses. Data in the table are presented as mean ± SE; means followed by different letters in the same row are significantly different between treatments, determined using the paired bootstrap test (*p* < 0.05). Standard errors were estimated using 5000 bootstraps. All life stages of *O. communa* were raised at the control temperature of 28 °C; experimental temperatures of 40 °C, 42 °C, or 44 °C were imposed from 12:00 to 15:00 daily to different life stages of *O. communa* placed in respective environmental chambers until a generation completed.

**Table 2 insects-13-00821-t002:** Effects of periodical high-temperature exposure on the fecundity of adult ragweed beetle *Ophraella communa*.

Population Parameters	Temperature (°C)			
	28 °C (*n*)	40 °C (*n*)	42 °C (*n*)	44 °C (*n*)
Oviposition days	24.96 ± 0.70 ^a^ (312)	35.53 ± 0.71 ^b^ (495)	44.39 ± 1.65 ^b^ (103)	23.15 ± 1.69 ^a^ (95)
Fecundity (offspring/female)	637.5 ± 12.97 ^b^ (312)	940.1 ± 19.37 ^c^ (495)	1059.8 ± 41.32 ^d^ (103)	480.0 ± 41.33 ^a^ (95)

The number of beetles is given in parentheses. Data in the table are represented as mean ± SE; means followed by different letters in the same row are significantly different between treatments, determined using the paired bootstrap test (*p* < 0.05). Standard errors were estimated using 5000 bootstraps. All life stages of *O. communa* were raised at the control temperature of 28 °C; experimental temperatures of 40 °C, 42 °C, or 44 °C were imposed from 12:00 to 15:00 daily to different life stages of *O. communa* placed in respective environmental chambers, until a generation completed.

**Table 3 insects-13-00821-t003:** Effects of periodical high-temperature exposure on the survival rates and sex ratio of different life stages of the ragweed beetle *Ophraella communa*.

Developmental Stage	Survival Rate and Sex Ratio (%)			
	28 °C (*n*)	40 °C (*n*)	42 °C (*n*)	44 °C (*n*)
Egg	91.88 ± 0.88 ^a^ (936)	80.00 ± 0.73 ^b^ (3020)	55.75 ± 1.27 ^d^ (1575)	73.82 ± 0.90 ^c^ (2311)
Larva	77.69 ± 1.42 ^a^ (668)	43.88 ± 1.01 ^c^ (1060)	52.28 ± 1.69 ^b^ (459)	38.69 ± 1.18 ^d^ (660)
Pupa	93.71 ± 0.94 ^a^ (626)	92.45 ± 0.82 ^a^ (980)	40.96 ± 2.30 ^b^ (188)	25.45 ± 1.67 ^c^ (168)
Pre-adult	66.88 ± 1.59 ^a^ (936)	32.45 ± 0.85 ^b^ (3020)	11.94 ± 0.81 ^c^ (1575)	7.27 ± 0.54 ^d^ (2311)
Sex ratio (female-to-male)	0.99 ± 0.08 ^a^ (936)	1.02 ± 0.07 ^a^ (3020)	1.21 ± 0.18 ^a^ (1575)	1.30 ± 0.21 ^a^ (2311)

The number of beetles is given in parentheses. Data in the table are represented as mean ± SE; means followed by different letters in the same row are significantly different between treatments, determined using the paired bootstrap test (*p* < 0.05). Standard errors were estimated using 5000 bootstraps. All life stages of *O. communa* were raised at the control temperature of 28 °C; experimental temperatures of 40 °C, 42 °C, or 44 °C were imposed from 12:00 to 15:00 daily to different life stages of *O. communa* placed in respective environmental chambers until a generation completed.

**Table 4 insects-13-00821-t004:** Effects of periodical high-temperature exposure on calculated population parameters for the ragweed beetle *Ophraella communa*.

Population Parameters	Temperature (℃)			
	28 °C	40 °C	42 °C	44 °C
*r* (d^−1^)	0.181 ± 0.002 ^a^	0.125 ± 0.001 ^b^	0.098 ± 0.003 ^c^	0.067 ± 0.003 ^d^
*λ* (d^−1^)	1.199 ± 0.003 ^a^	1.134 ± 0.002 ^b^	1.103 ± 0.003 ^c^	1.070 ± 0.003 ^d^
*R* _o_	212.50 ± 10.69 ^a^	154.08 ± 7.07 ^b^	69.31± 7.06 ^c^	19.73 ± 2.62 ^d^
*T* (d)	29.58 ± 0.24 ^a^	40.16 ± 0.23 ^b^	43.45 ± 0.61 ^c^	44.37 ± 0.19 ^d^

*r*: intrinsic rate of increase; *λ*: finite rate of increase; *R*_0_: net reproductive rate; *T*: mean generation time (offspring/individual). Standard errors (SE) were estimated using the bootstrap technique with 5000 re-samplings; differences between treatments were compared using a paired bootstrap test implemented with TWOSEX-MSChart. The means in the same row followed by different lowercase letters indicate significant differences between treatments (*p* < 0.05).

## Data Availability

The datasets used and analyzed in this study are available from the corresponding author upon request.

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
