# Peer review of "Effects of Fluctuating Thermal Regimes on Life History Parameters and Body Size of Ophraella communa"

_insects, 2022, doi:10.3390/insects13090821_

Round 1
Reviewer 1 Report
In this study, Zhao et al. analyzed the effects of fluctuating thermal regimes on life history parameters and body size of Ophraella communa. The story is interesting and the paper is well documented. I recommend to accept it after some light reversions. There are several comments for this manuscript:
Minor comments:
1. Abstract: In my opinion, you should mention more of your results not just two sentences from line 17 to 21.
2. line 14-15: This sentence needs to be rephrased.
3. line 34: The ragweed leaf beetle Ophraella communa…
4. line 36: Also as known that O. communa is an….
5. line 53: change “measured” to “analyzed/investigated”
6. line 59: change “Ambrosia artemisiifolia” to “A. artemisiifolia”
7. line 370 and 374: the reference formats are different. Correct!
8. Line 377: D. simulans should be italicized.
9. Reference format should be rechecked.
Author Response
In this study, Zhao et al. analyzed the effects of fluctuating thermal regimes on life history parameters and body size of Ophraella communa. The story is interesting and the paper is well documented. I recommend to accept it after some light reversions. There are several comments for this manuscript:
Dear Professor:
Thank you very much for the nice comments. We would like to express our sincere thanks to you for the constructive and positive comments.
We have revised the manuscript, point by point responses to the comments are listed below this letter, which was marked in blue.
Minor comments:
- Abstract: In my opinion, you should mention more of your results not just two sentences from line 17 to 21.
Response: Thank you very much for your suggestion. It has been modified according to your suggestion.
- line 14-15: This sentence needs to be rephrased.
Response: Thank you very much for your suggestion. It has been modified according to your suggestion.
- line 34: The ragweed leaf beetle Ophraella communa…
Response: it has been added.
- line 36: Also as known that O. communa is an….
Response: Thanks for your advice. We had revised as suggested.
- line 53: change “measured” to “analyzed/investigated”
Response: Thanks for your advice. We had revised as suggested.
- line 59: change “Ambrosia artemisiifolia” to “A. artemisiifolia”
Response: Thanks for your advice. We had revised as suggested.
- line 370 and 374: the reference formats are different. Correct!
Response: Thanks for your advice. We had revised as suggested.
- Line 377: D. simulans should be italicized.
Response: Thanks for your advice. We had revised as suggested.
- Reference format should be rechecked
Response: Thanks for your advice. We had revised as suggested.
Reviewer 2 Report
The authors conducted a series of experiments aimed to estimate the influence of a short-term heat stress on various physiological parameters of Ophraella communa, a leaf beetle widely used for biological control of a noxious invasive plant, common ragweed. The results of the study are not only interesting for fundamental insect physiology but also have important applied value because they can be used to predict potential impact of possible further climate change on insect population dynamics. The experiments were well conducted, the primary analysis of the results (as far as I can understand) is correct. However, the presentation of some results is unclear and confusing. Besides, I would not agree with some of the conclusions made by the authors. Therefore, although the paper has merit and can be published, it undoubtedly needs numerous substantial changes: explanations, improvements and corrections (see my comments below).
Lines 17-23. The second half of the Abstract is rather confusing. First, several negative impacts of repeated exposure to high temperature on various biological parameters of Ophraella communa are listed: an increase in the duration of development, a decrease in fecundity etc. (lines 17-19) then mentioned two positive effects (on mating success and production of viable offspring) and (rather unexpectedly) concluded that “climate change is anticipated to impact this insect positively“ (lines 22-23). However, Figs 6 and 7 very clearly show that the total cumulative effect is undoubtedly negative: both individual adult fitness and the rate of population growth in all experimental treatments are lower than in controls. I think that “exposure to heat stress negatively affects ragweed beetles, but they were able to survive and reproduce” (lines 286-287) would be much more correct conclusion for the Abstract.
Besides, it is stated that “Our findings demonstrate that O. communa can rapidly adapt to climate change within a single generation” (lines 21-22) although in reality this adaptation was suggested by corresponding changes in only secondary important parameters such as egg hatchability which in the second generation is still negatively affected by high temperatures (Fig. 7).
Dear authors, please, consider, explain or correct the above confusing conclusions.
Line 53: I would replace “disrupted” with “influence” because “disrupted” means only negative effect which can’t be postulated “a priory’ before the study but should be demonstrated by the study.
Figure 1: As far as I understand, this figure is based not on original but on literature data. If yes, all appropriate references should be included in the legend (lines 89-90).
Table 1: Each term in the first column should be much better explained.
For example, not “Egg” but “ The time (or the duration) of egg development” etc. I guess that “Pre-adult” means the total duration of pre-adult development (at least it is equal to the sum of the first 3 rows). Then what is “Adult”? Is it adult longevity? If yes, what is the meaning of the next 3 rows also concerning longevity?
“Longevity” (the 8th row) seems to be the same as “Life expectancy” (lines 216-217 and Figure 4)? At least the values (52.87 etc.) are exactly the same... Please, clearly explain all terms and keep the same terms over whole text.
Is “Longevity of survived to adult” counted from the egg stage or from the last (adult) molt? Please, clearly explain how the values in all rows were calculated.
Generally, I would recommend to give the formula for calculation of each parameter.
Tables 3 and 4. As far as I understand, the method used was not “continuous high temperature exposure“ but short term (3 h) high temperature pulses. If yes, please, correct the head of this table.
Figure 7c: in the legend (line 267) F2 egg hatchability is indicated, whereas along the Y-axis of this figure F1 egg hatchability is shown. Please, correct either figure or legend.
Lines 382-383: I would not agree with this conclusion. Table 3 shows that the negative effect of heat exposure on the survival rate was far from being “moderate”: for example, in the treatment with +44 C the total survival was about 10% of that in controls. Once again, I think that “exposure to heat stress negatively affects ragweed beetles, but they were able to survive and reproduce” (lines 286-287) would be much more exact final general conclusion.
Round 2
Reviewer 2 Report
The manuscript was substantially improved. Now it can be published.